# Can process mapping and a multisite Delphi of perioperative professionals inform our understanding of system-wide factors that may impact operative risk?

Daniel Stubbs [iD],[1,2] Tom Bashford [iD],[1,3] Fay Gilder,[4] Basil Nourallah,[5] Ari Ercole [iD],[2] Nicholas Levy,[5] John Clarkson[1]

[1]Healthcare Design Group, Department of Engineering, University of Cambridge, Cambridge, UK
[2]Division of Anaesthesia, University of Cambridge Department of Medicine, Cambridge, UK
[3]Department of Anaesthesia, Cambridge University Hospitals NHS Foundation Trust, Cambridge, UK
[4]Princess Alexandra Hospital NHS Trust, Harlow, UK
[5]Department of Anaesthesia, West Suffolk Hospital, Bury Saint Edmunds, UK

**Correspondence to**
Dr Daniel Stubbs;
djs225@cam.ac.uk

## ABSTRACT

**Objectives** To examine whether the use of process mapping and a multidisciplinary Delphi can identify potential contributors to perioperative risk. We hypothesised that this approach may identify factors not represented in common perioperative risk tools and give insights of use to future research in this area.

**Design** Multidisciplinary, modified Delphi study.

**Setting** Two centres (one tertiary, one secondary) in the UK during 2020 amidst coronavirus pressures.

**Participants** 91 stakeholders from 23 professional groups involved in the perioperative care of older patients. Key stakeholder groups were identified via process mapping of local perioperative care pathways.

**Results** Response rate ranged from 51% in round 1 to 19% in round 3. After round 1, free text suggestions from the panel were combined with variables identified from perioperative risk scores. This yielded a total of 410 variables that were voted on in subsequent rounds. Including new suggestions from round two, 468/519 (90%) of the statements presented to the panel reached a consensus decision by the end of round 3. Identified risk factors included patient-level factors (such as ethnicity and socioeconomic status), and organisational or process factors related to the individual hospital (such as policies, staffing and organisational culture). 66/160 (41%) of the new suggestions did not feature in systematic reviews of perioperative risk scores or key process indicators. No factor categorised as 'organisational' is currently present in any perioperative risk score.

**Conclusions** Through process mapping and a modified Delphi we gained insights into additional factors that may contribute to perioperative risk. Many were absent from currently used risk stratification scores. These results enable an appreciation of the contextual limitations of currently used risk tools and could support future research into the generation of more holistic data sets for the development of perioperative risk assessment tools.

## INTRODUCTION

Understanding, predicting and communicating risk is fundamental to perioperative practice.[1] The use of surgical risk stratification tools is a cornerstone of programmes such as the National Emergency Laparotomy

STRENGTHS AND LIMITATIONS OF THIS STUDY
⇒ Novel use of process mapping to identify key perioperative stakeholders.
⇒ Multidisciplinary Delphi panel to gain breadth of perspective.
⇒ Performed across two sites.
⇒ Comprehensive results may be of use to other researchers designing perioperative research databases.
⇒ Results may be limited by low response rate in final round (although majority of consensus decisions made in round 2).

Audit (NELA).[2] When using the output of such tools in day-to-day practice it is important to remember that they are developed from external electronic data sets and their application to decisions at the level of an individual centre or patient is complex and fraught with difficulty.[3] Risk scores commonly consist of patient and surgical factors that are associated with relevant outcomes.[4] However, performance of such scores is far from perfect.[5] This raises the question as to whether currently unmeasured factors may improve our understanding and prognostication of risk. There is strong evidence that this is the case. Unwarranted variation is widespread within the National Health Service, with between-centre differences leading to discrepancies in cost, efficiency and, most importantly, patient outcome.[6] Significantly, organisational or system factors have been shown in a recent analysis of NELA data to be associated with worse outcomes.[7] However, such factors do not feature in commonly used risk assessment tools

Healthcare is increasingly digitised with electronic health records (EHRs) capturing detailed events from across the hospital system. Perioperatively, EHRs may hold data pertaining to an individual's baseline state,

operation details, physiological responses under anaesthesia and perioperative complications. EHRs therefore appear to offer an appealing substrate to identify and test factors associated with perioperative outcome. In reality, due to the complexity of modern healthcare, the data they hold do not accurately capture the 'true' patient state but the record is in fact biased by the care processes involved in the recording of such data.[8] To identify nascent factors that may broaden our understanding of perioperative risk, including at the level of the healthcare system, it is thus vital that we understand how that care is delivered, and the affect this might have on the electronic record.

A system may be defined as a collection of elements serving a common purpose, with emergent behaviour arising through their interaction. Such systems may be considered as simple, complicated or complex, with complex systems exhibiting certain key behaviours such as non-linearity, non-scalability and emergence.[9] Healthcare has been posited as such a complex system, which, perioperatively, may include preassessment clinics, operating theatres, staff from multiple disciplines, patients, equipment, consumables and care processes. All of these may generate data which could provide novel insights into our understanding of risk. Researchers in other healthcare settings, where similar causal pathways may exist, have already recognised that standard statistical techniques may not be comprehensive enough to interrogate and understand such systems.[10] Therefore, there is a need to consider additional strategies such as those used to understand other complex systems.[11 12]

Our aim in this paper was to employ a systems approach[11] to develop a holistic data set that a breadth of perioperative professionals felt captured all dimensions of perioperative risk. The output of such an endeavour would enable a rational approach to extracting information from EHRs for future research and enable a clearer understanding of how these data are captured as part of the wider care process. Such strategies, as well as offering a clearer scientific rationale for future hypothesis testing, could also encourage better data governance—by only extracting data fields from the EHR that were felt to be of clear importance.

To develop our consensus data set, we employed a stepwise approach. First, we sought to visualise how perioperative care is delivered through process mapping. This is a technique which enables complex systems to be visualised as a series of steps that has been used in various healthcare settings.[13 14] These process maps were then used to identify the range of perioperative professionals involved in patient care, and whose views we needed to capture. Using this list of 'stakeholders' we next conducted a modified Delphi[15] across two hospital sites to gain consensus on the breadth of factors at both patient, operation and system levels that were felt to impact on patient outcome.

## METHODS

### Setting and approvals

Participants were recruited from two UK hospital trusts using EHRs. Cambridge University Hospitals Trust (CUH) is a tertiary referral centre offering secondary and tertiary-level surgical services to patients across the East of England. The West Suffolk Hospital (WSH) is a district general hospital based in Bury St Edmunds. WSH offers a range of secondary care services while referring patients for tertiary care to specialist centres including CUH. This study forms part of the 'Designing Improved Surgical Care for Older people' (DISCO) study. DISCO is jointly sponsored by the University of Cambridge and CUH.

### Methodological approach and techniques

Systems engineers use a variety of techniques to interrogate and design complex systems. A framework for applying these tools to healthcare has been recently jointly published by the UK's Royal Academy of Engineers, Royal College of Physicians and Academy of Medical Sciences.[11] In this study, we used brainstorming interviews and graphical elicitation,[16] process mapping and a Delphi process to ensure we captured the views of professionals (stakeholders) from across the perioperative system in two distinct sites. A project flow diagram demonstrating the sequential use of these techniques is shown in figure 1.

### Stakeholder identification and process mapping

We formed a local steering group of experienced perioperative professionals. The group consisted of a consultant anaesthetist, geriatrician and surgeon alongside a senior physiotherapist, occupational therapist, matron and operations manager. Brainstorming interviews were conducted to identify stakeholder groups who should be represented on the Delphi panel. Interviews were structured around the iteration of process maps representing a stereotypical 'high-risk' surgical patient undergoing vascular surgery. Vascular surgery was chosen due to the ability to draw comparisons between elective and emergency cases as well as the need for clinical input from a range of perioperative professionals. Stakeholders were identified from these maps and then chosen for representation on the Delphi panel if at least one member of the steering group felt this was appropriate.

### Delphi panel formation

Representatives were approached from each stakeholder group across both sites aiming for a complementary spread of subspecialties between sites. Individuals were approached by lead researchers, provided with written information and gave informed consent prior to each Delphi round.

### Delphi structure: round 1

The Delphi consisted of three rounds and was distributed using an online survey tool (Qualtrics—www.qualtrics.com). In round 1, individuals were asked to provide free text suggestions on what they felt *'Could contribute*

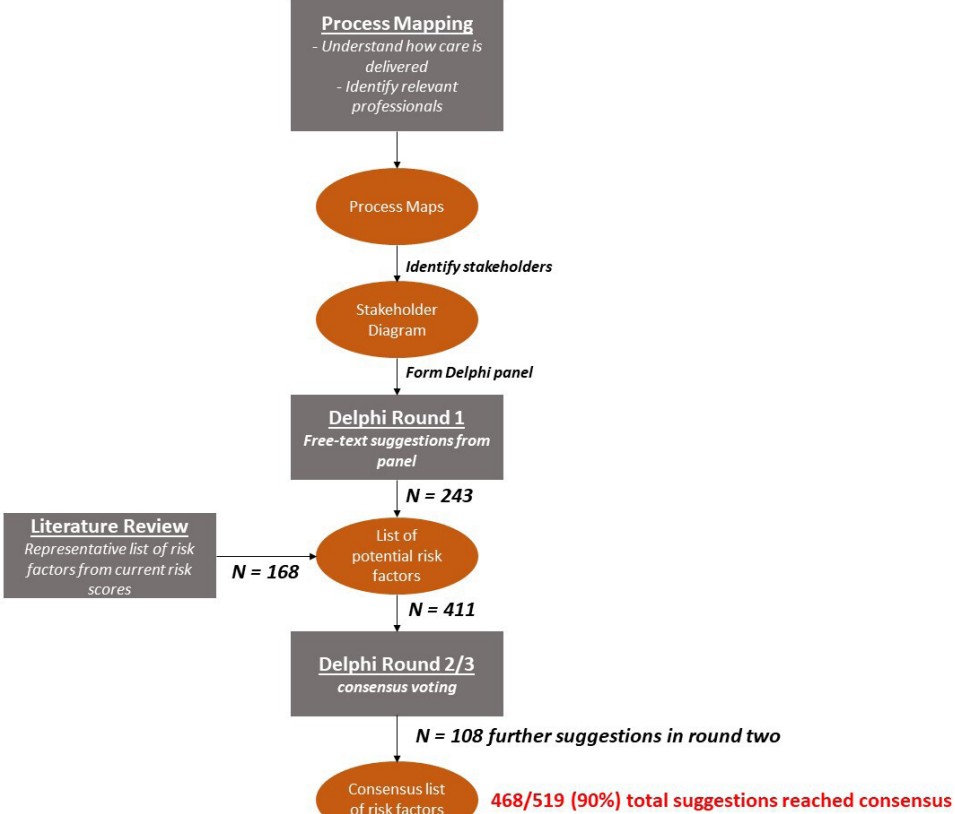

**Figure 1** Project stages. Process mapping was performed with a steering group of experienced professionals, with maps iterated between brainstorming interviews. At the conclusion of this stage two outputs were generated—the maps themselves and a stakeholder diagram showing the viewpoints we sought to capture in our Delphi. The Delphi was conducted across three rounds with the first seeking free text suggestions from the panel. These were combined with risk factors featuring in a systematic review of common perioperative risk tools.[4] All suggestions were then voted on by the panel across two further rounds to gain consensus. Our final consensus list of risk factors is available in online supplemental file 2. N=number of risk factors gained from each source.

*to a poor-outcome in an older patient undergoing surgery'.* We defined a poor outcome to be where an individual lost their independence after surgery or suffered a complication (such as a myocardial infarction). Suggestions from the panel were combined with known important risk factors identified from a systematic review of perioperative risk scores.[4] The full list of scores is shown in online supplemental file 1. All risk factors (literature and panel) were voted on in the second and third rounds using a 5-point Likert scale. Participants could also provide free text comments and clarifications. A minority of questions in round 2 asked for specific free text responses to expand on suggestions or provide relevant cut-offs (eg, frailty score thresholds). New suggestions from round 2 were included for voting in round 3. Before each major group of suggestions (eg, 'comorbidities') respondents were asked to indicate on the same 5-point Likert scale how 'measurable' suggestions within this category were felt to be. This was to gauge the feelings of the panel on the practicality of this information being captured within an electronic record.

### Delphi structure: rounds 2 and 3

All potential risk factors were presented for voting in a hierarchy consisting of groups and subgroups reflecting free text suggestions. There were three broad domains— patient level (*that may have existed prior to an individual's current admission: eg, comorbidities*), admission level (*the circumstances and events occurring in any given admission*) and system level (*suggestions that were related to the structure or running of a service within a healthcare organisation*). Tracking of free text data for generating questionnaires was conducted using ATLAS.ti (www.atlasti.com) with quantitative analysis conducted in R V.3.6.3.[17]

### Definition of consensus

Consensus was defined using criteria modified from a Delphi of quality indicators for patients with traumatic brain injury.[18] Given the relative heterogeneity of our panel, consensus for a given question required at least 50% of respondents to address it, a median score of >3.5 and an IQR of ≤1. Consensus exclusion at the end of round 2 was stricter, requiring a score of <2.5, an IQR of ≤1 and no scores of 5 ('very important').

### Comparing to common literature sources

At the conclusion of the Delphi, all novel and consensus suggestions were compared with a systematic review of perioperative structure and process indicators,[19] as well as

whether they had been examined in the statistical development of each risk score.[20–33] This approach was chosen to ensure that our final list of factors contained a solid core of important patient-level factors while allowing us to critique novel findings against a comprehensive literature source.

## Groupings, definitions and results reporting

Certain questions in the Delphi allowed participants to vote on cut-points (eg, '*a "high-risk" body mass index (BMI) is above 30*') or on an overarching concept that could encompass multiple factors (for instance, 'complications' could encompass 'perioperative myocardial infarction' as well as surgical complications). For transparency, all questions are presented in the online supplemental file 2 and included in relevant numerators or denominators within the Results section. However, these cut-offs were not considered when reporting on factors present in literature risk scores unless they were explicitly mentioned (eg, a risk score defined a specific cut-off).

## Patient and public involvement in project

The protocol for our project was reviewed by the CUH patient and public involvement panel in November 2017 and received favourable responses. Our project aims are clearly aligned with the James Lind Association priority setting partnership conducted with the National Institute of Academic Anaesthesia.[34]

## RESULTS
## System mapping

Process maps representing the care of an elective and emergency vascular surgical patient were generated based on an understanding of local care pathways (abridged version in figure 2, full examples in online supplemental file 1—figures 1 and 2). These diagrams were revised during brainstorming interviews with our steering group. Maps were used to identify relevant stakeholders across the perioperative pathway. In total, 52 unique staff groups were identified. Of these, 33 were nominated by at least one member of the steering group for representation on the Delphi panel (figure 3).

## Stakeholders and Delphi participants

Invitations to participate were sent to 91 professionals, identified by leads in both trusts (63 from CUH, 28 from WSH). These covered 23 broad professional groups as well as subspecialty expertise. Numbers of representatives are shown in online supplemental file 1—figure 3. Participants were able to contribute views from more than one perspective (eg, anaesthetist AND intensive care doctor). Response rates ranged from 51% (n=46) in round 1 to 19% (n=17) in round 3. Conduct was significantly impacted by the coronavirus pandemic—distribution of the second and third rounds was delayed by 6 months due to wave 1 and the final round was completed as case

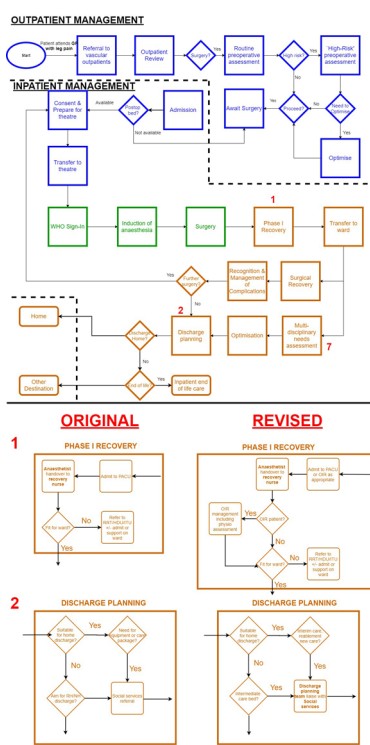

**Figure 2** Simplified process map of a patient referred for elective vascular surgery at our institution. *Blue* boxes highlight preoperative care, *green* intraoperative care, *orange* postoperative care. Boxes indicate processes, diamonds indicate decision points. Dashed lines indicate boundary between community and hospital. Stakeholders (eg, anaesthetist) are shown in bold text. Lower panel highlights more granular view of numbered processes including revisions to initial diagram following discussions with a steering group of perioperative professionals. Fully granular maps for elective and emergency pathways are available in the online supplemental file 1. OIR, overnight intensive recovery (level 2 postanaesthesia care unit (PACU) for high-risk patients). GP, General Practitioner; HDU, High Dependency Unit; ITU, Intensive Therapy Uniit; NH, Nursing Home; RH, Residential Home; RRT, Rapid Response Team (Outreach).

pressures built prior to the second national UK lockdown in November 2020.

## Minimum data set: variables from the literature

Twenty-five risk scores were identified from a 2013 systematic review.[4] A full list of scores and their component variables are demonstrated in online supplemental file 1—table 1 and figure 4. Any of these variables not suggested in round 1 were included by default for voting in rounds 2 and 3.

## Delphi round 1

From the literature, 168 variables (representing specific measurements or characteristics) were identified (online supplemental file 1—figure 4). From free text responses from participants, 411 suggestions for variables were identified, including 243 unique or refined definitions from those present in the literature. Eighty of the 168 (48%)

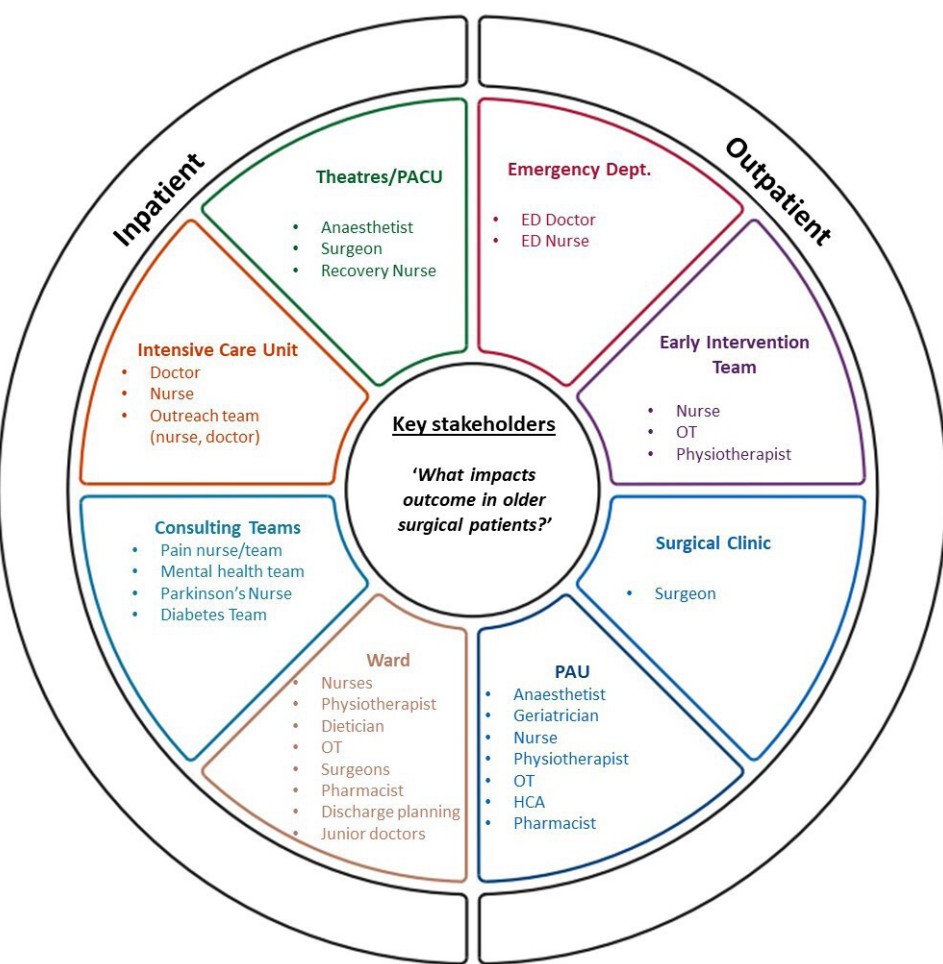

**Figure 3** Stakeholder diagram. Stakeholder groups identified for representation on a Delphi panel to understand what defines a high-risk surgical patient and impacts on their outcome. All groups were identified by a collaborative process mapping exercise with a steering group of key perioperative professionals. Diagram template adapted from the 'Improving Improvement Toolkit' available at www.iitoolkit.com, used with permission.[48] ED, emergency department; HCA, healthcare assistant; OT, occupational therapist; PACU, postanaesthesia care unit; PAU, preassessment unit.

literature variables were not suggested by the panel in round 1.

All suggestions were grouped into a framework separating out suggestions pertaining to the *patient*, their *admission* and the *organisation* caring for them. For clarity of questioning, suggestions were further organised into related groups (such as comorbidities), subgroups (eg, cardiovascular comorbidities) and then a granular level that represented specific concepts or definitions (eg, 'electrocardiogram changes' and 'left bundle branch block'). This structure is demonstrated in figure 4. Distribution of new suggestions across these domains of *patient*, *admission* and *organisation* is shown in figure 5A. None of the suggestions that appeared to pertain to health system organisation or performance featured in currently used risk scores (figure 5A).

### Delphi rounds 2 and 3

In round 2, participants were able to vote on all levels of the questionnaire hierarchy (figure 4) including groups and subgroups. In total, 409 suggestions including 17 groups, 34 subgroups and 358 variables (of which 117

were operationalised definitions of a variable—such as 'left bundle branch block' representing an 'important ECG change') were voted on. Two suggestions from the risk scores (cut-off values for defining polypharmacy) were held until round 3 as free text suggestions for this threshold were sought from panellists in round 2. Full details of all questions that were voted on were presented for voting across the final two rounds and are available in online supplemental file 2, with a summary shown in table 1.

Having determined in round 2 that chronological age met the consensus criteria for inclusion, participants in round 3 were asked to suggest age thresholds that might be used to indicate a higher risk 'older' patient. From 19 participants, 16 gave a numerical threshold with the most common being 70 or older (n=6, 38%). Two participants highlighted that they felt an appropriate threshold would vary with the type of surgery and one, that they did not believe chronological age was a valid criterion.

At the conclusion of round 2, three hundred and fifty-seven (87%) of statements reached consensus criteria

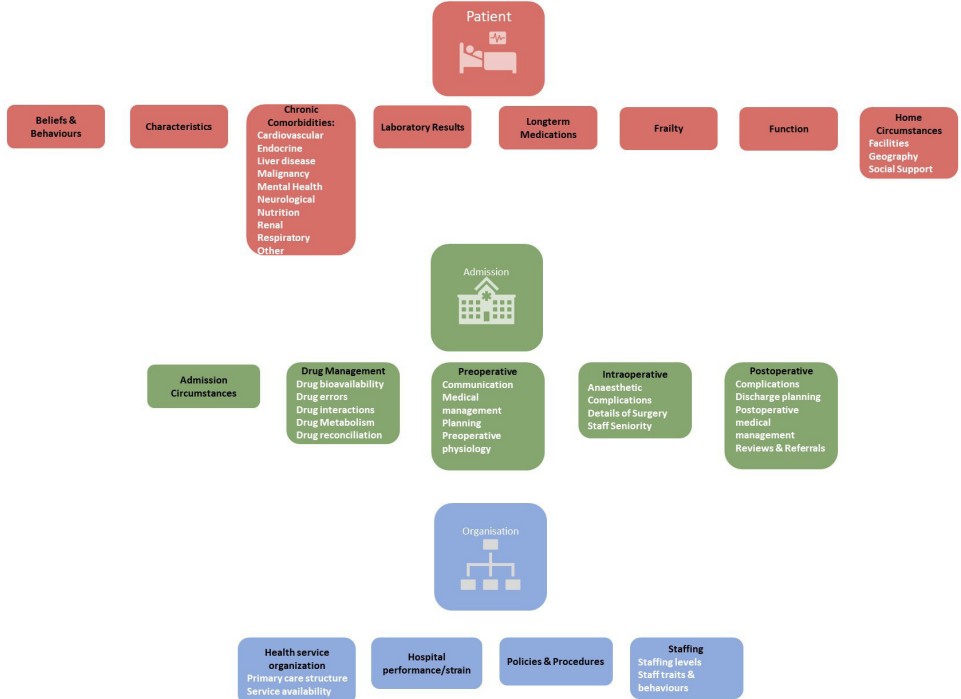

**Figure 4** Delphi structure showing main risk factor groups identified through participant suggestions and the literature. The questionnaire enabled people to vote on three broad domains (patient, admission, organisation) and relevant groups (black text in lozenges). Many of these groups had relevant subgroups (white text). Participants could also vote on individual data elements (variables or their definitions) within each of these. Full list of these is available in the online supplemental file 2.

for inclusion. Analysis of free text suggestions identified a further 108 suggested variables for voting in the third Delphi round.

The median score for measurability in round 2 was 4 (IQR: 3–4). Laboratory results were felt to be the most measurable (attracting a median score of 5) with patient *beliefs and behaviours, health service organisation, hospital performance and strain,* and *policies and procedures* attracting median scores of 3.

In the third Delphi round, 110 of 158 (70%) variables met the consensus criteria for inclusion, one variable ('*shortness of breath on strenuous exercise*') met the criteria for exclusion. In total, across both rounds, 468 of 519 (90%) suggestions that were presented to the panel reached consensus decisions (figure 5B). In round 3, median measurability was put at 4 (IQR: 4–5) with only *beliefs and behaviours* and *health service organisation* attracting median scores of less than 4. Across both latter rounds all of the suggestions encompassing health system organisation reached the definition for consensus.

### Comparison to literature variables
To assess whether variables might have been previously examined but excluded from published risk scores due to a lack of statistical significance the original papers describing each were examined.[20–33] Twenty-five (7%) of the 351 unique suggestions from rounds 1 and 2 had been previously examined at an earlier phase in the development of at least one risk tool (details in online supplemental file 2).

All novel suggestions were also compared against a list of process and structure indicators from a 2018 systematic review.[19] One hundred and eleven of 352 (32%) novel suggestions could be mapped against one or more metrics identified in this paper. When new suggestions defining patient-level characteristics (eg, comorbidities) were excluded, 88 of 161 (54%) new suggestions relating to admission circumstances or organisational function were represented. Variables that did not appear included markers of system performance (eg, *number of vacant posts, delayed transfer of care rates*) and staffing (eg, *occupational therapy cover*) as well as examples of postoperative complications (eg, *anastomotic breakdown*). Full details can be seen in table 2 in online supplemental file 1 and the online supplemental file 2.

### DISCUSSION
This study demonstrates the views of multidisciplinary clinicians on factors felt to influence perioperative risk. We feel that both our findings and methods will be of interest to researchers, clinicians, data scientists and managers. Our results highlighted that factors related to in-hospital events, organisational structure and hospital performance were felt to be important. Such suggestions were conspicuously absent from commonly used perioperative risk scores (figure 5A)[4] but are compatible with work demonstrating intercentre variation in outcome.[7] Our final list of variables (available in online supplemental file 2, summarised in table 1) is likely to be of use

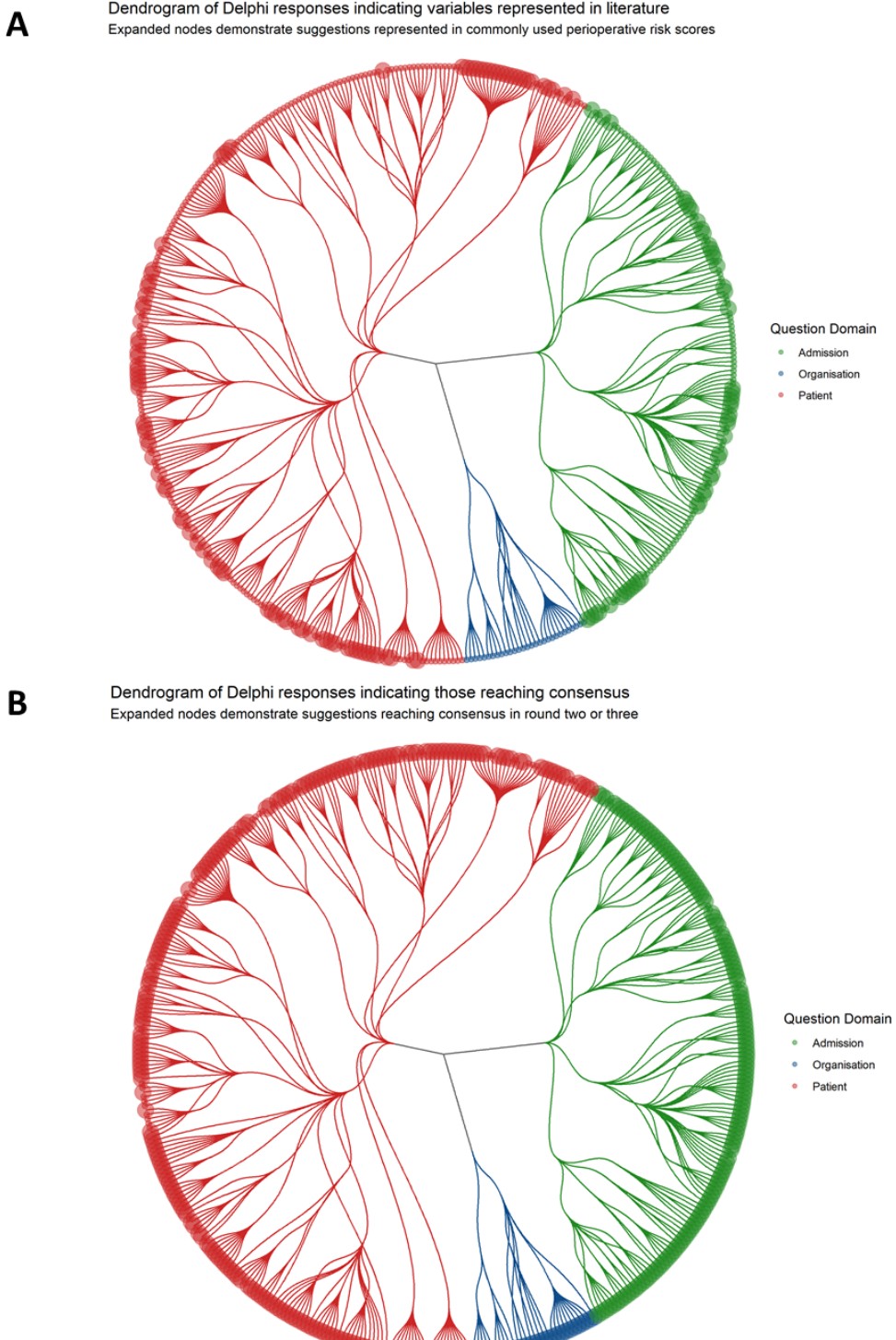

**A** Dendrogram of Delphi responses indicating variables represented in literature
Expanded nodes demonstrate suggestions represented in commonly used perioperative risk scores

Question Domain
- Admission
- Organisation
- Patient

**B** Dendrogram of Delphi responses indicating those reaching consensus
Expanded nodes demonstrate suggestions reaching consensus in round two or three

Question Domain
- Admission
- Organisation
- Patient

**Figure 5** Representation of the Delphi questionnaires as a dendrogram, demonstrating hierarchical organisation of questions. Colours represent broad domains of statements being voted on by the panel; pertaining to a patient, circumstances of admission or those pertaining to the organisation caring for them. Final questions that were voted on are represented by terminal leaves. Branch points represent questionnaire structure. In panel A, expanded nodes represent those variables derived from a systematic review of commonly used perioperative risk scores. In panel B, expanded nodes represent those that reached criteria for consensus inclusion in a data set seeking to understand drivers of perioperative risk in older people.

to researchers in the field seeking to intelligently curate data from their own EHRs.

We hypothesised that process mapping may enable us to identify a panel of stakeholders whose expertise captured all facets of perioperative care and, that in doing so, we may gain novel insights. This approach is at least partially vindicated in that most factors voted on (351/519—68%) were suggested by the panel rather than in our selected

**Table 1** Summary of findings from a multisite Delphi of perioperative professionals to identify variables of importance in determining or predicting outcome after surgery

| Domain | Group | Example variables | n | n (R2) | n (R3) | Consensus: any (n (%)) |
|---|---|---|---|---|---|---|
| Admission | Analgesic plan | Analgesic plan \| Multimodal analgesia \| Opioid deprescribing predischarge \| Opioid dose \| Regional/epidural | 5 | 5 | 0 | 5 (100) |
| Admission | Circumstances | Chest injuries \| Delayed presentation \| Emergency (vs elective) admission \| Neck of femur fracture \| Sepsis or organ failure on admission | 12 | 8 | 4 | 11 (92) |
| Admission | Details of surgery | BUPA classification \| Dutch Surgical Society Score \| Indication for surgery (pathology) \| Intraperitoneal, aortic or intrathoracic operation \| Length of surgery | 15 | 11 | 1 | 12 (80) |
| Admission | Discharge planning | Delayed transfer of care \| Discharge planning \| Needing care on discharge | 3 | 3 | 0 | 3 (100) |
| Admission | Drug management | Absence of a specific route of administration (eg, oral) \| Antiemetics and Parkinson's medications \| Cytochrome P450 inhibitors/inducers \| Drug errors \| Timing of drug reconciliation (eg, by pharmacy) | 21 | 16 | 5 | 21 (100) |
| Admission | Intraoperative | Details of surgery \| Intraoperative blood loss \| Intraoperative $CO_2$ control \| Night-time operating \| Operator fatigue | 27 | 22 | 4 | 26 (96) |
| Admission | Postoperative | Location and level of care \| Multidisciplinary working \| Number of attachments \| Time to mobilisation \| Ward moves/handovers | 14 | 14 | 0 | 14 (100) |
| Admission | Postoperative complications | Anastomotic breakdown \| Blood transfusion \| Cardiovascular complications \| Delirium \| Fall | 34 | 31 | 2 | 33 (97) |
| Admission | Preoperative | Appropriate preoperative ward \| DNR status and escalation plan \| How long patients expect to stay in hospital \| Multidisciplinary working and communication \|Use of a risk tool | 25 | 22 | 3 | 25 (100) |
| Admission | Preoperative physiology | Preoperative acid base status (eg, pH, bicarbonate) \| Preoperative acute renal impairment \| Preoperative CVS impairment \| Preoperative delirium \| Preoperative electrolytes | 24 | 24 | 0 | 24 (100) |
| Organisation | Health services organisation | Availability of services (eg, community therapy) \| Practice type \| Presence of a telephone advice service \| Primary care structure \| Tertiary admission/transfer | 7 | 5 | 2 | 7 (100) |
| Organisation | Hospital performance | Bed occupancy rates \| Delayed transfer of care (DToC) rates \| ED admission wait times \| Intensive care occupancy \| Number of vacant posts | 11 | 4 | 6 | 10 (91) |
| Organisation | Policies and procedures | Care pathway documents \| Enhanced recovery programmes \| Procedures for minimising conflicts between teams \| Procurement | 6 | 5 | 1 | 6 (100) |
| Organisation | Staffing | Empathy \| Enthusiasm and engagement \| Expertise in the care of older people \| Nursing cover \| Training levels | 13 | 11 | 2 | 13 (100) |
| Patient | Beliefs and behaviours | Adoption of sick role \| Dependency on staff \| Engagement with advice/treatment \| Medication compliance \| Patient motivation | 12 | 9 | 2 | 11 (92) |
| Patient | Characteristics | Age \| ASA score \| Ethnicity \| Genetics \| Hearing or visual loss \| Quality of life | 10 | 7 | 1 | 8 (80) |
| Patient | Comorbidities | (Fast) AF \| BMI \| Chronic dysrhythmia \| Diabetes \| Symptoms that limit eating | 168 | 99 | 48 | 146 (87) |

Continued

**Table 1** Continued

| Domain | Group | Example variables | n | n (R2) | n (R3) | Consensus: any (n (%)) |
|---|---|---|---|---|---|---|
| Patient | Frailty | Deconditioning \| Frailty score \| Frailty syndrome: delirium \| Frailty syndrome: falls \| Frailty syndrome: incontinence \| Loss of muscle | 19 | 6 | 7 | 13 (68) |
| Patient | Function | Able to drive \| Able to use stairs \| Continent \| Defined by ADLs \| Independent with transfers and walking | 17 | 14 | 2 | 16 (94) |
| Patient | Home circumstances | Access difficulties (eg, steps) \| Accommodation type \| Bathroom access \| Bedroom access \| Can the house support single-storey living? | 40 | 22 | 15 | 37 (93) |
| Patient | Laboratory results | Albumin \| ALT \| APTT \| AST \| Bilirubin \| Cholesterol \| CK \| Glucose \| Haematocrit \| Haemoglobin \| Laboratory results \| Lymphocytes \| Nucleated RBC \| Platelets \| PT \| Thrombocytopenia \| Transferrin \| Transthyretin \| Triglycerides \| WCC | 21 | 13 | 2 | 15 (71) |
| Patient | Long-term medications | Antiplatelets \| Diuretics \| Direct oral anticoagulants (DOACs) \| Immunosuppressants \| Number of medications \| Warfarin | 15 | 7 | 5 | 12 (80) |

Domain and Group refer to section of Delphi questionnaire; a subset of example variables considered within these sections are shown for clarity (full details available in online supplemental file 2). n=total number of variables/thresholds considered by participants in each section. n(Rx)=number reaching consensus inclusion in round 2 or 3. Consensus (any)=number reaching consensus (%) across any round. ADLs = Activities of Daily Living ADLs, Activities of Daily Living; AF, atrial fibrillation; ALT, alanine transferase; APTT, activated partial thromboplastin time; ASA, American Society of Anesthesiologists; AST, aspartate transferase; BMI, body mass index; BUPA, British United Provident Association; CK, creatine kinase; CVS, cardiovascular system; DNR, do not resuscitate; ED, emergency department; PT, prothrombin time; RBC, red blood cell; WCC, white cell count.

subset of risk scores. We acknowledge that it is possible that some of our suggestions may feature in other risk scores. We required a sufficiently broad baseline to assure the validity of our results while critiquing the scale of any novel insights. Given the rising popularity of risk scores, we did however require a pragmatic approach. Therefore, we selected scores (including those commonly used in UK practice such as P-POSSUM the Portsmouth modification of the Physiological and Operative Severity Score for the enUmeration of Mortality and morbidity)[4] that had been validated in distinct surgical populations.

The importance of 'non-patient'-level factors in prognostication has not been thoroughly explored, perhaps reflecting the challenges in recording such concepts electronically. When considering quantitative data from EHRs, improved risk assessment can be seen by incorporating implicit information such as the timing of blood samples, as well as their resulting value.[35] Here, this timing information is presumably recognising a potentially otherwise undocumented recognition of a clinically unwell patient by a diligent clinician taking, and interpreting, blood samples out of hours. The results of this study suggest that variables not captured in a hospital EHR, such as detailed aspects of social circumstances, may be important. A machine learning model containing only 'sociomarkers' (derived from place of residence) has yielded comparable performance in the prediction of childhood asthma exacerbation compared with the use of patient-level data.[36] However, conflicting findings have

been shown in other settings. Although qualitative data derived from patient interviews have suggested that social support and psychological state are influences on individual readmissions with heart failure,[37][38] small studies looking to incorporate questionnaire-derived markers have failed to demonstrate improved prognostication[39] despite the apparent use of 'unstructured' free text data in another study.[40] Perioperatively, it is well known that clinician judgement improves the performance of commonly used risk tools such as Surgical Outcome Risk Tool.[41] What is unclear is what additional factors this clinician judgement encapsulates and whether it may reflect an appreciation or implicit judgement of some of the factors identified in this study.

We feel that a strength of our study has been to identify avenues for future research, by providing a comprehensive list of social and health system factors that researchers may want to explore in future data sets. The documented differences in outcome between institution,[41] socioeconomic strata[42] and our findings mean that exploring these factors warrants further investigation. One approach to try and address this may be to broaden the definition as to what we view as constituting 'healthcare data'. The Institute of Medicine suggests that relevant ancillary data sources may include human resources records and patient complaints.[43] This could conceivably aid with the incorporation of factors such as staffing level, with complaints data, if adequately coded using tools such as the Healthcare Complaints Analysis Tool, identifying

poor staff–patient relationships and communication.[44] However, we acknowledge the challenges (both logistical and governance) in formulating linked data sets capable of answering these questions, especially at significant scale.

However, investigation of these factors may yield crucial insights. First, for predictive analyses, an appreciation of system factors in influencing patient-level risk could aid in reducing uncertainty around calculated predictions, while providing crucial contextual information for shared decision-making. Increasing the breadth of future data sets obviously raises the need to ensure that developed predictive models are suitably parsimonious. Second, a comprehensive data set containing these novel variables could form a substrate for causally focused analyses. When conducting such an analysis, the generation of a causal model should proceed in a data-naïve manner, to ensure adequate recognition of latent variables with potential causal effects.[45] Our results could be a starting point for such work, by highlighting variables of potential causal importance that would be missed with data-driven approaches on currently existing data sets.

This study took the unique step of drawing on techniques from systems engineering to structure a Delphi survey of professionals. Our methods will be of direct relevance to those involved in informatics or quality improvement work. A systems approach is a way of addressing problems holistically, aware of the interaction between elements and subsequent unexpected behaviour.[11 46] Suggestions from the panel related to the circumstances of admission, hospital performance and external pressures. A conceptual framework is thus that the ultimate outcome of any patient stay depends on the interaction between factors at different levels. Conceptually, this hierarchical agreement is appealing, with the importance of external factors, such as coronavirus pressures, a key part of current daily practice. Beyond our unique methodological approach and multidisciplinary nature of our results, a further point of originality within our work is that the Delphi panel were specifically asked to suggest factors that could result in a loss of independence on discharge, a key concern of patients undergoing surgery.[47] Despite this, it is not widely considered in perioperative risk stratification.[4]

We do, however, acknowledge the falloff in response rate across rounds. This arose due to the significant pressures of the coronavirus pandemic and that many participants on our panel were undertaking additional clinical duties. We would counter this by highlighting the breadth of specialties surveyed, and that the response rate in round 2 was 38%, and that a majority of consensus factors (357/490) were identified in this round. It is possible that the low response rate in round 3 skewed the remaining factors towards reaching consensus, but this should make our resultant list of variables especially sensitive (although potentially less specific). The high rates of consensus could also reflect true strength of feeling but

may have arisen due to issues with questionnaire length, panel composition and response rates.

## CONCLUSION

This study demonstrates the feasibility of using systems engineering tools to identify and engage clinicians in identifying factors felt to impact on patient-relevant outcomes after surgery. The results themselves highlight that these professionals identify non-patient-level factors as modulators of perioperative risk. Further work is needed to prioritise these results to develop electronic surrogates to validate their significance in real data sets.

**Contributors** DS: conception, Delphi design, participant recruitment, analysis, primary manuscript drafting, corresponding author, approval of final version, agreement to be accountable for all aspects of the work (guarantor). JC and AE: conception, Delphi design, critical revisions to manuscript, approval of final version, agreement to be accountable for all aspects of the work. TB: analytical advice, Delphi revisions, critical revisions to manuscript, approval of final version, agreement to be accountable for all aspects of the work. FG: stakeholder identification, critical revisions to manuscript, approval of final version, agreement to be accountable for all aspects of the work. NL: site-specific recruitment and set-up, manuscript drafting, critical revisions to manuscript, approval of final version, agreement to be accountable for all aspects of the work. BN: site-specific recruitment, critical revisions to manuscript, approval of final version, agreement to be accountable for all aspects of the work. AE: conception, Delphi design, critical revisions to the manuscript, approval of final version, agreement to be accountable for all aspects of the work.

**Funding** This research was funded, in whole or in part, by the Wellcome Trust (grant number: 220542/Z/20/Z) (to DS). This research was supported by the NIHR Cambridge Biomedical Centre (BRC 1215 20014).

**Disclaimer** The views and opinions expressed by the authors in this publication are those of the authors and do not necessarily reflect those of the NHS, the NIHR or the Department of Health.

**Competing interests** None declared.

**Patient and public involvement** Patients and/or the public were involved in the design, or conduct, or reporting, or dissemination plans of this research. Refer to the Methods section for further details.

**Patient consent for publication** Not applicable.

**Ethics approval** This study involves human participants and was approved by the London and Surrey Borders Research Ethics Committee (reference: 19/LO/1648). Participants gave informed consent to participate in the study before taking part.

**Provenance and peer review** Not commissioned; externally peer reviewed.

**Data availability statement** All data relevant to the study are included in the article or uploaded as supplementary information.

**ORCID iDs**
Daniel Stubbs http://orcid.org/0000-0003-2778-5226
Tom Bashford http://orcid.org/0000-0003-0228-9779

Ari Ercole http://orcid.org/0000-0001-8350-8093

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
