## [Reviewer comments · BMJ Open]

ARTICLE DETAILS

TITLE (PROVISIONAL)	Can process mapping and a multi-site Delphi of perioperative professionals inform our understanding of system-wide factors that may impact operative risk?
AUTHORS	Stubbs, Daniel; Bashford, Tom; Gilder, Fay; Nourallah, Basil; Ercole, Ari; Levy, Nicholas; Clarkson, John

VERSION 1 – REVIEW

REVIEWER	Meguid, Robert University of Colorado at Denver - Anschutz Medical Campus, Surgery
REVIEW RETURNED	02-Jun-2022

GENERAL COMMENTS	Thank you for the privilege of re-reviewing “Can process mapping and a multi-site Delphi of perioperative professionals inform our understanding of system-wide factors that may impact operative risk? by Stubbs, et al. This is a novel study engaging a large number of broadly representative stakeholders in perioperative patient care. It is aspirational in some respects, regarding what data we should collect in the future. It is well-written. Overall, this is a useful contribution to the literature on perioperative risk stratification tools. With significant revisions, I think this will be of interest to the readers of BMJ, or a surgical journal. I have the following comments: The paper starts out leading the reader to believe that we will be shown the list of variables that should be collected in order to develop better surgical risk assessment tools. However, by the end, the authors conclude that this is a proof-of-concept study that demonstrates use of process mapping and modified Delphi techniques can be successfully used to obtain input from a broad group of clinicians. The actual variables identified by the study are provided in supplemental material, which I suspect most readers will not access. The paper seems to be more of a techniques paper than presenting results of the study. I recommend improving the clarity of the message to either a techniques paper, or more prominently displaying the findings of the 3 rounds of Delphi panels by moving supplemental table 2 into the body of the manuscript as a main table, and condense it by themes of the elements currently presented in the table. The study identifies risk factors for adverse outcomes after surgery in “older” patients. How is “older” defined? The selection of risk tools seems arbitrary and omits the major risk assessment tools used in the United States, where I practice surgery and medicine. These include the American College of Surgeons Surgical Risk Calculator, SURPAS (Surgical Risk Preoperative
---

	Assessment System), and the Emergency Surgery Score (ESS). These have varying degrees of parsimony of required predictor variables. Why didn't the authors take these into account? The authors need to address the limitations of actualization of the findings of their study. Currently, there are no datasets (at least that I am aware of) which contain all of the potential predictor variables identified in this study. Therefore, models incorporating them can not be built and tested. They do mention that the "non-patient level factors" identified by the participants "as modulators of perioperative risk" have not been studied. Thus, the significance of these is unclear, and I question why resources should be directed to amass these into a large (multicentered or international) database if their importance is unclear. Can the authors provide literature to support the importance of these variables, and thus the cost of developing tools to collect them? The authors should mention the pros and cons of using an expansive list of potential risk predictors for model development as compared to a parsimonious approach to risk model development as supported by the large body of literature on SURPAS (Surgical Risk Preoperative Assessment System). The references need editing, including adding dates accessed for websites (Ref 2 and 33 are missing this). Ref 22, 25, 26, 27, 28 and 41 appear incomplete. Reference is made to a study on "traumatic brain injury" with reference number 50, which I can't find in the reference list (ends at 44). The figure labeling is confusing – there are two figures labeled "figure 3" in addition to "supplemental figure 3." Figures 1 and 2 aren't labeled.
--	---

REVIEWER	Liu, Lihua Chinese PLA General Hospital, Institute of Hospital Management Research
REVIEW RETURNED	21-Jun-2022

GENERAL COMMENTS	The research did a meaningful work that provided additive evidence for evaluating operative risk. However, when creating a risk-score in the reality, some useful and meaningful structural factors may prove little effects quantificationally and data collection may be another big problem. The study was presented at great length. I wonder whether it would be better if it could follow a guidance on conducting and reporting delphi studies.
--

VERSION 1 – AUTHOR RESPONSE

Reviewer: 1

Thank you for the privilege of re-reviewing "Can process mapping and a multi-site Delphi of perioperative professionals inform our understanding of system-wide factors that may impact operative risk?" by Stubbs, et al. This is a novel study engaging a large number of broadly representative stakeholders in perioperative patient care. It is aspirational in some respects, regarding what data we should collect in the future. It is well-written. Overall, this is a useful contribution to the literature on perioperative risk stratification tools. With significant revisions, I think this will be of interest to the readers of BMJ, or a surgical journal. I have the following comments:

The paper starts out leading the reader to believe that we will be shown the list of variables that should be collected in order to develop better surgical risk assessment tools. However, by the end, the authors conclude that this is a proof-of-concept study that demonstrates use of process mapping and modified Delphi techniques can be successfully used to obtain input from a broad group of clinicians. The actual variables identified by the study are provided in supplemental material, which I suspect most readers will not access. The paper seems to be more of a techniques paper than presenting results of the study. I recommend improving the clarity of the message to either a techniques paper, or more prominently displaying the findings of the 3 rounds of Delphi panels by moving supplemental table 2 into the body of the manuscript as a main table, and condense it by themes of the elements currently presented in the table.

We thank the reviewer for the suggestion to summarise our results in the main text. We have done this by inserting a new Table 1 that summarises the number of variables participants voted on within each major subdivision and the number (and percentage) reaching consensus by round. Due to the number of suggestions (within comorbidities especially) the table became unwieldy (4-5 pages +) when all component variables were included, therefore we have displayed 'example' variables from each section (apart from lab results where the use of acronyms allows a maintenance of clarity), with the reader referred to supplementary if they want further details. We are happy to revisit this decision with both the reviewer and editor if they feel a larger table is appropriate.

The study identifies risk factors for adverse outcomes after surgery in "older" patients. How is "older" defined?

We asked participants to suggest (in free-text) specific age thresholds that they felt would denote an 'older' person as being at higher risk. We have reviewed all of these responses and included a new paragraph in the results section (under 'Delphi rounds two and three') that includes a summary of these responses.

The selection of risk tools seems arbitrary and omits the major risk assessment tools used in the United States, where I practice surgery and medicine. These include the American College of Surgeons Surgical Risk Calculator, SURPAS (Surgical Risk Preoperative Assessment System), and the Emergency Surgery Score (ESS). These have varying degrees of parsimony of required predictor variables. Why didn't the authors take these into account?

We recognise that the number of risk stratification tools in practice is dramatically increasing with the growth of available data sources. We required a mechanism to select a meaningful 'core' of variables to critique the suggestions of our panel against and to ensure that a reasonable breadth of demonstrably important variables were presented for voting. The risk scores deconstructed for this purpose were drawn from a comprehensive systematic review published by Moonesinghe et al that specifically identified scores that had been validated as meaningful in heterogeneous surgical cohorts. Whilst we recognise that this is not exhaustive, we felt it should provide a meaningfully representative sample, capture scores commonly used in UK practice (e.g. P-POSSUM). However, we have amended our discussion to highlight this potential limitation.

The authors need to address the limitations of actualization of the findings of their study. Currently, there are no datasets (at least that I am aware of) which contain all of the potential predictor variables identified in this study. Therefore, models incorporating them can not be built and tested. They do mention that the "non-patient level factors" identified by the participants "as modulators of perioperative risk" have not been studied. Thus, the significance of these is unclear, and I question why resources should be directed to amass these into a large (multicentered or international) database if their importance is unclear. Can the authors provide literature to support the importance of

these variables, and thus the cost of developing tools to collect them?

We agree that the implications of many of these variables (for prognostic or causal modelling purposes) are, as yet, unproven. Although certain of these factors (say those pertaining to individual hospital performance – such as bed state or staff vacancies) are conceivably available in separate hospital databases we are not currently aware of a systematic approach to combine these with patient level variables for analysis.

However, we feel strongly that this ‘absence of evidence’ should not be constituted to assume that efforts to examine their importance should not be considered, and have amended our discussion to emphasise this point. Studies clearly demonstrate between-centre variation in surgical outcome (e.g. Oliver et al Br J Anesthes 2018). Our findings could also be used for causal inference studies, where an emphasis on devising a causal model without reference to available data is advocated (e.g. Tennant et al Int J Epidemiol 2021).

We have therefore reworked our discussion to both recognise the scale of the challenge of this problem but also to highlight the potential opportunities for future research.

The authors should mention the pros and cons of using an expansive list of potential risk predictors for model development as compared to a parsimonious approach to risk model development as supported by the large body of literature on SURPAS (Surgical Risk Preoperative Assessment System).

We have noted the importance of ensuring new predictive models are suitably parsimonious.

The references need editing, including adding dates accessed for websites (Ref 2 and 33 are missing this). Ref 22, 25, 26, 27, 28 and 41 appear incomplete. Reference is made to a study on “traumatic brain injury” with reference number 50, which I can’t find in the reference list (ends at 44).

We apologise for these issues with referencing and have ensured that these are now appropriately labelled and complete

The figure labeling is confusing – there are two figures labeled “figure 3” in addition to “supplemental figure 3.” Figures 1 and 2 aren’t labeled.

We apologise for this and have now corrected all figure legends

Reviewer: 2

Dr. Lihua Liu, Chinese PLA General Hospital, Chinese PLA General Hospital

Comments to the Author:

The research did a meaningful work that provided additive evidence for evaluating operative risk. However, when creating a risk-score in the reality, some useful and meaningful structural factors may prove little effects quantificationally and data collection may be another big problem. The study was presented at great length. I wonder whether it would be better if it could follow a guidance on conducting and reporting delphi studies.

We thank the reviewer for their comments on our manuscript and we agree that the importance of novel factors identified in this work remain unproven (including the scale of their quantitative effect). However as detailed in our response to reviewer 1 we view this as a strength of our paper, by posing challenging hypotheses that could be explored in future work. We have however reworked our

discussion to emphasise the challenges of this process and the uncertainty of the results.

We also acknowledge that our combination of methods is complex and does create challenges with article length. We have tried to make the article more readable and signpost key sections and made several changes for brevity (see general edits). Whilst we acknowledge a standardised reporting framework would be useful, to our knowledge no formal checklist exists yet for consensus studies. Indeed, the importance of this has been noted and work is underway to develop such a guideline (see: <https://researchintegrityjournal.biomedcentral.com/articles/10.1186/s41073-022-00122-0>) which will be of great benefit to the field.